# FUSION OVER THE GRASSMANNIAN FOR HIGH-RANK MATRIX COMPLETION

## ABSTRACT

This paper presents a new paradigm to cluster and complete data lying in a union of subspaces using points in the Grassmannian as proxies. Our main theoretical contribution exploits the geometry of this Riemannian manifold to obtain local convergence guarantees. Furthermore, our approach does not require prior knowledge of the number of subspaces, is naturally suited to handle noise, and only requires an upper bound on the subspaces' dimensions. We detail clustering, completion, model selection, and sketching techniques that can be used in practice. We complement our analysis with synthetic and real-data experiments, which show that our approach performs comparable to the state-of-the-art in the *easy* cases (high sampling rates), and significantly better in the *difficult* cases (low sampling rates), thus shortening the gap towards the fundamental sampling limit of HRMC.

## 1 INTRODUCTION

This paper focuses on the problem known as *high-rank matrix completion* (HRMC) Eriksson et al. (2012); Elhamifar (2016), which can be summarized as follows: given a subset of entries in a data matrix $\mathbf{X}$ whose columns lie near a union of subspaces, our goals are to $(i)$ complete the unobserved entries, $(ii)$ cluster the columns according to the subspaces, and $(iii)$ learn the underlying subspaces. HRMC is motivated by a wide range of applications, including tracking moving objects in computer vision Cao et al. (2016); Chen & Lerman (2009); Ho et al. (2003); Hong et al. (2006); Huang et al. (2004); Kanatani (2001); Lu & Vidal (2006), predicting target interactions for drug discovery Gan (2013); Malhat et al. (2014); Mashayekh et al. (2019); Perualila-Tan et al. (2016), identifying groups in recommender systems Koohi & Kiani (2017); Ullah et al. (2014); Zhang et al. (2021), and learning network toppologies Eriksson et al. (2011). However, existing methods generally lack supporting theory Balzano et al. (2012a); Lane et al. (2019a); Yang et al. (2015); Pimentel et al. (2014); Pimentel-Alarcón et al. (2016); Elhamifar (2016); Tsakiris & Vidal (2018), yield uninterpretable results that are unsuitable for scientific and security settings Ji et al. (2017); Peng et al. (2020), and require many more samples than the information-theoretic limit suggests Pimentel-Alarcon & Nowak (2016).

The fundamental difficulty behind all these shortcomings lies in assessing distances (e.g., euclidean, or in the form of inner products) between partially observed vectors. The simple reason is that computing distances requires overlapping observations, which become increasingly unlikely in low-sampling regimes Eriksson et al. (2012). To circumvent this problem, we introduce a new paradigm to cluster incomplete vectors, using subspaces as proxies, thus avoiding the need to calculate distances or inner products or other notions of similarity between incomplete vectors, as other methods require. Our main idea is to assign each (incomplete-data) point to its own (full-data) subspace, and simultaneously minimize over the Grassmann manifold: (a) the chordal distance between each point and its assigned subspace, to guarantee that the subspace stays *near* the observed entries, and (b) the geodesics between subspaces of all data, to encourage the subspaces from points that belong together to *fuse* (i.e, represent the same space). This optimization is performed over the Grassmann manifold, rather than over Euclidean space. At the end of this minimization, we proceed to cluster the proxy subspaces using standard procedures like k-means or spectral clustering Bottou & Bengio (1995); Elkan (2003); Sculley (2010); Tong et al. (2020); Veenstra et al. (2016); Von Luxburg (2007); Wen (2020). The clustering of the proxy subspaces gives in turn a clustering for the incomplete-data (goal $ii$). After clustering, the missing entries can be filled (goal $i$) using any standard low-rank matrix completion algorithm Recht (2011a). Finally, once the data is clustered and completed, the underlying subspaces can be trivially inferred (goal $iii$) with a singular value decomposition.

Our optimization over the Grassmannian brings two key advantages. First and foremost, it gives us the ability to cluster incomplete-data through their subspace proxies. Secondly, our approach enables us to obtain local convergence guarantees using well-known manifold optimization results. Taking advantage of this machinery we are able to show that our Riemannian gradient descent steps will converge to a critical point, which is usually the strongest attainable guarantee for non-convex formulations of this sort Pimentel et al. (2014); Pimentel-Alarcón et al. (2016). Additionally, our approach does not require prior knowledge of the number of subspaces, is naturally suited to handle noise, and only requires an upper bound on the subspaces' dimension. We complement our theoretical results with experiments on both synthetic and real data. These experiments show that even on this first *vanilla* version, (a) our approach performs comparable to the state-of-the-art in the *easy* cases (high sampling rates), and (b) our approach significantly outperforms the state-of-the-art in the *difficult* cases (low sampling rates), thus shortening the gap towards the fundamental sampling limit of HRMC (see Figures 4-5). Besides establishing a new the state-of-the-art, this introductory version of our algorithm intends to show the potential of our new paradigm. We hope that this publication spurs new insights and interest that leads to more sophisticated variants of our formulation that ultimately result in a polynomial-time algorithm that can provably attain the information-theoretic sampling limit Pimentel-Alarcon & Nowak (2016).

The rest of the paper is organized as follows. Section 2 presents our main contributions, namely, our formulation, our Riemannian gradient directions, and our convergence guarantees. Section 3 summarizes previous work and its relation to our contributions. Section 4 discusses special considerations that need to be considered when applying our methodology in practice. All our experiments are in Section 5. Section 6 discusses existing challenges and potential improvements of our formulation, which will be the focus of our future work. Section 7 summarizes our contributions and conclusions. In the interest of self-containment, all the fundamental concepts and background necessary to understand the methods of this paper are summarized in Appendices B-E. Appendix F includes the derivations of our gradient steps, Appendix G outlines related work required for the proof of our main result, and Appendix A includes an explicit example of the HRMC setup and goals. We hope that this example will give insight and intuition to the unfamiliar reader, and at the same time show the subtleties and difficulty of the HRMC problem.

## 2 MODEL AND MAIN RESULTS

Let $\mathbf{x}_1, \ldots, \mathbf{x}_n \in \mathbb{R}^m$ lie near a union of subspaces with dimension upper bounded by $r$. Let $\mathbf{x}_i^\Omega \in \mathbb{R}^{|\Omega_i|}$ denote the observed entries of $\mathbf{x}_i$, indexed by $\Omega_i \subset \{1, \ldots, m\}$. We propose assigning to each observed vector $\mathbf{x}_i^\Omega$ a proxy subspace $\mathbb{U}_i := \text{span}(\mathbf{U}_i)$. Our goal is to estimate the true subspace $\mathbb{U}_i^\star$ to which $\mathbf{x}_i$ belongs by (a) enforcing that the proxy space $\mathbb{U}_i$ contains a possible completion of $\mathbf{x}_i^\Omega$ and (b) minimizing the distance between individual proxy spaces $\mathbb{U}_i$ and $\mathbb{U}_j$ to build consensus. This is done via the following optimization problem, where the first term achieves goal (a) and the second term achieves goal (b):

$$\min_{\mathbf{U}_1, \ldots, \mathbf{U}_n \in \mathbb{S}(m,r)} \quad \sum_{i=1}^n d_c^2(\mathbf{x}_i^\Omega, \mathbf{U}_i) \; + \; \frac{\lambda}{2} \sum_{i,j=1}^n d_g^2(\mathbf{U}_i, \mathbf{U}_j), \tag{1}$$

where

$$d_c(\mathbf{x}_i^\Omega, \mathbf{U}_i) := \sqrt{1 - \sigma_1^2(\mathbf{X}_i^{0\mathsf{T}} \mathbf{U}_i)} \qquad \text{and} \qquad d_g(\mathbf{U}_i, \mathbf{U}_j) := \sqrt{\sum_{\ell=1}^r \arccos^2 \sigma_\ell(\mathbf{U}_i^\mathsf{T} \mathbf{U}_j)}.$$

Here $\lambda \geq 0$ is a regularization parameter, $\sigma_\ell(\cdot)$ denotes the $\ell^{\text{th}}$ largest singular value, and $\mathbf{X}_i^0$ is the orthonormal matrix spanning all the possible completions of a non-zero $\mathbf{x}_i^\Omega$. The space of all possible completions of $\mathbf{x}_i^\Omega$ is therefore $\mathbb{X}_i^0 := \text{span}(\mathbf{X}_i^0)$, which clearly contains the true data $\mathbf{x}_i$. The matrix $\mathbf{X}_i^0$ can be easily constructed as follows. If $\mathbf{x}_i^\Omega = \mathbf{0}$, then $\mathbf{X}_i^0 = \mathbf{I}$, the identity matrix. Otherwise, $\mathbf{X}_i^0$ is the $m \times (m - |\Omega_i| + 1)$ matrix formed with $\mathbf{x}_i^\Omega$ normalized and filled with zeros in the unobserved rows, concatenated with the $(m - |\Omega_i|)$ canonical vectors indicating the unobserved rows of $\mathbf{x}_i^\Omega$. For

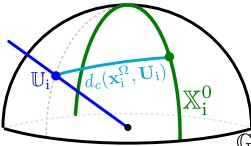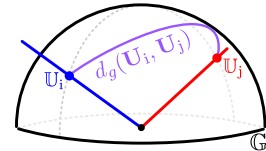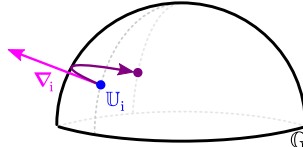

Figure 1: The semi-spheres represent the Grassmannian $\mathbb{G}(m, r)$, where each point $\mathbb{U}_i$ represents a subspace (in the particular case of $\mathbb{G}(3, 1)$, the line going from the origin to $\mathbb{U}_i$). **Left:** Intuitively, the *chordal* distance $d_c(\mathbf{x}_i^\Omega, \mathbf{U}_i)$ is an informal measure of distance between the subspace $\mathbb{U}_i$ and an incomplete point $\mathbf{x}_i^\Omega$. The left image should only be taken as intuition since $\mathbb{X}_i^0$ may not live on the same Grassmannian and the chordal distance should not be thought of as a geodesic distance. **Center:** The *geodesic* distance $d_g(\mathbf{U}_i, \mathbf{U}_j)$ measures the distance over the Grassmannian between $\mathbb{U}_i$ and $\mathbb{U}_j$. **Right:** The Euclidean gradient vector $\boldsymbol{\nabla}_i$ *falls* out of the Grassmann manifold; to account for the Grassmannian curvature, each geodesic step needs to be adjusted according to (4).

example, if $\mathbf{x}_i^\Omega \neq \mathbf{0}$ is observed in the first $|\Omega_i|$ rows, then

$$
\mathbf{X}_i^0 = \left[ \begin{array}{c|c} \frac{\mathbf{x}_i^\Omega}{\|\mathbf{x}_i^\Omega\|} & \mathbf{0} \\ \hline \mathbf{0} & \mathbf{I} \end{array} \right] \begin{array}{l} \left.\vphantom{\frac{x}{x}}\right\} |\Omega_i| \\ \\ \left.\vphantom{\frac{x}{x}}\right\} m - |\Omega_i|. \end{array}
$$

$$\underbrace{\phantom{XXXXXXXXXX}}_{m - |\Omega_i| + 1}$$

When $\mathbf{x}_i^\Omega$ is fully observed, $\mathbf{X}_i^0$ simplifies to $\mathbf{x}_i$ normalized. Recall that Grassmannians $\mathbb{G}(m, r)$ are quotient spaces of Stiefel manifolds $\mathbb{S}(m, r)$ by action of the orthogonal group of $r \times r$ orthonormal matrices. Since both terms $d_c(\mathbf{x}_i^\Omega, \mathbf{U}_i)$ and $d_g(\mathbf{U}_i, \mathbf{U}_j)$ are invariant under this quotient, the objective function in (1) does not depend on the choice of basis, and descends to a function on the Grassmannian.

**Why should this work?** The *chordal distance* $d_c(\mathbf{x}_i^\Omega, \mathbf{U}_i)$, as defined in Wei Dai & Milenkovic (2012), is not a formal distance on the Grassmannian, but rather measures how far $\mathbb{U}_i$ is from containing a possible completion of $\mathbf{x}_i^\Omega$. More precisely, $d_c(\mathbf{x}_i^\Omega, \mathbf{U}_i)$ is the cosine of the angle between the nearest completion of $\mathbf{x}_i^\Omega$ and the $r$-plane $\mathbb{U}_i$. If the top singular value $\sigma_1(\mathbf{X}_i^{0\top}\mathbf{U}_i)$ is 1, then $\mathbb{X}_i^0$ and $\mathbb{U}_i$ intersect on at least a line, meaning that the proxy space $\mathbb{U}_i$ contains a possible completion of $\mathbf{x}_i^\Omega$. While merely forcing $\mathbb{U}_i$ to contain a possible completion offers no way to distinguish one possible completion to another, consensus among data is built as different proxies $\mathbb{U}_i$ and $\mathbb{U}_j$ are forced towards one another by the geodesic term. In other words, $\mathbb{U}_i$ and $\mathbb{U}_j$ are allowed to be near each other, and hence form clusters, only if they both contain possible completions of both $\mathbf{x}_i^\Omega$ and $\mathbf{x}_j^\Omega$. The term *chordal distance* used in this way is adopted from Wei Dai & Milenkovic (2012) and should not be confused with the more common chordal distance between points on the Grassmannian Conway et al. (1996). See Figure 1 to build some intuition.

**Solving** (1) The gradients of (1) with respect to $\mathbf{U}_i$ over the Grassmannian are given by:

$$
\boldsymbol{\nabla} d_c^2(\mathbf{x}_i^\Omega, \mathbf{U}_i) = -2\sigma_1(\mathbf{X}_i^{0\top}\mathbf{U}_i)(\mathbf{I} - \mathbf{U}_i\mathbf{U}_i^\top)\mathbf{v}_i\mathbf{w}_i^\top, \tag{2}
$$

$$
\boldsymbol{\nabla} d_g^2(\mathbf{U}_i, \mathbf{U}_j) = -\sum_{\ell=1}^r \frac{2\arccos\sigma_\ell(\mathbf{U}_i^\top\mathbf{U}_j)}{\sqrt{1 - \sigma_\ell^2(\mathbf{U}_i^\top\mathbf{U}_j)}} \cdot (\mathbf{I} - \mathbf{U}_i\mathbf{U}_i^\top)\mathbf{v}_{ij}^\ell\mathbf{w}_{ij}^{\ell\top}, \tag{3}
$$

where $\mathbf{v}_i$ and $\mathbf{w}_i$ are the leading left and right singular vectors of $\mathbf{X}_i^0\mathbf{X}_i^{0\top}\mathbf{U}_i$, and $\mathbf{v}_{ij}^\ell$, $\mathbf{w}_{ij}^\ell$ are the $\ell^{\text{th}}$ left and right singular vectors of $\mathbf{U}_j\mathbf{U}_j^\top\mathbf{U}_i$. The key behind these expressions is that tangent vectors on the Grassmannian can be computed as projections of gradient vectors in Euclidean space Absil et al. (2009); Edelman et al. (1998). In fact, that is exactly what the gradient expressions in (2) and (3) are: $-2\sigma_1(\mathbf{X}_i^{0\top}\mathbf{U}_i)\mathbf{v}_i\mathbf{w}_i^\top$ in (2) is the gradient of $d_c^2(\mathbf{x}_i^\Omega, \mathbf{U}_i)$ with respect to the entries of $\mathbf{U}_i$ in Euclidean space. The multiplication by $\mathbf{I} - \mathbf{U}_i\mathbf{U}_i^\top$ takes the horizontal direction of the tangent vector with respect to the quotient, thus mapping the gradient from Euclidean space to the Grassmannian. The same is true for the term $\mathbf{I} - \mathbf{U}_i\mathbf{U}_i^\top$ in (3). To summarize, (2) and (3) are gradient directions in

the manifold of subspaces, rather than matrices: (2) is the steepest direction along which the subspace $\mathbb{U}_i$ can be descended to a potential subspace that contains $\mathbf{x}_i^\Omega$, and (3) is the steepest direction along which the subspace $\mathbb{U}_i$ can be descended to the subspace $\mathbb{U}_j$. The derivations of these gradients are in Appendix F. Putting together the chordal and geodesic gradients, our overall descent direction for $\mathbf{U}_i$ is given by

$$\boldsymbol{\nabla}_i \; := \; \boldsymbol{\nabla} d_c^2(\mathbf{x}_i^\Omega, \mathbf{U}_i) + \frac{\lambda}{2} \sum_{j=1}^n \boldsymbol{\nabla} d_g^2(\mathbf{U}_i, \mathbf{U}_j).$$

Observe that $\mathbf{U}_i - \eta \boldsymbol{\nabla}_i$ *falls* out of the Grassmannian for every step size $\eta \neq 0$ (see Figure 1). To adjust for the curvature of the manifold, the update after taking a geodesic step of size $\eta$ over the Grassmannian in the direction of $-\boldsymbol{\nabla}_i$ is given by equation (2.65) in Edelman et al. (1998), which in our context reduces to:

$$\mathbf{U}_i \; \leftarrow \; \begin{bmatrix} \mathbf{U}_i \mathbf{E}_i & \boldsymbol{\Gamma}_i \end{bmatrix} \begin{bmatrix} \operatorname{diag} \cos(\eta \boldsymbol{\Upsilon}_i) \\ \operatorname{diag} \sin(\eta \boldsymbol{\Upsilon}_i) \end{bmatrix} \mathbf{E}_i^\mathsf{T}, \tag{4}$$

where $\boldsymbol{\Gamma}_i \boldsymbol{\Upsilon}_i \mathbf{E}_i^\mathsf{T}$ is the compact singular value decomposition of $-\boldsymbol{\nabla}_i$. In our implementation, we use Armijo step sizes, given by $\eta = \beta^\nu \eta_0$, where $\eta_0 > 0$, and $\beta, \gamma \in (0, 1)$ are the Armijo tuning parameters related to the initial step size and step search granularity Absil et al. (2009), and $\nu$ is the smallest non-negative integer such that

$$\sum_{i=1}^n f_i(\mathbf{U}_i) - f_i(R_{\mathbf{U}_i}(\beta^\nu \eta_0 \boldsymbol{\nabla}_i)) \; \geq \; -\gamma \sum_{i=1}^n \langle \boldsymbol{\nabla}_i, \beta^\nu \eta_0(-\boldsymbol{\nabla}_i) \rangle,$$

where $f_i(\mathbf{U}_i)$ is the component of the objective function (1) holding i fixed and ranging over j, and $R_{\mathbf{U}_i}(\boldsymbol{\Delta})$ performs the geodesic step described by (4) in the direction of $\boldsymbol{\Delta}$.

**Convergence guarantees.** One advantage of our approach is that we can use standard techniques as in Mishra et al. (2019) to obtain local convergence guarantees like the following:

**Theorem 2.1.** *Let $\{(\mathbf{U}_1, \mathbf{U}_2, \ldots, \mathbf{U}_n)\}$ be a sequence of iterates generated by the geodesic steps given by equation* (4) *with Armijo steps sizes $\eta$ as defined above. Then the sequence will converge to a critical point of* (1)*.*

*Proof.* It suffices to show the (rather technical) fact that the gradient steps in (4) are an instance of Accelerated Line Search (ALS) given in Absil et al. (2009) and outlined in Algorithm 1 in Appendix G, where the product manifold $\mathbb{G}^n$ will serve as the Riemannian manifold $\mathcal{M}$, the tangent space of which is the cartesian product of tangent spaces of each constituent $\mathbb{G}$. To see this, let $T\mathcal{M}$ denote the *tangent bundle* of (set of all tangent vectors to) $\mathcal{M}$, and let $T_\mathbb{U}\mathcal{M}$ denote the tangent space of $\mathcal{M}$ at $\mathbb{U} \in \mathcal{M}$. In our case, $\mathbb{U}$ is the tuple $(\mathbf{U}_1, \ldots, \mathbf{U}_n)$, and equation (4) serves as the *retraction* $R_{\mathbf{U}_i}$ on each component, so that $R_\mathbb{U} = (R_{\mathbf{U}_1}, \ldots, R_{\mathbf{U}_n})$. One can verify that this is indeed a retraction by recognizing (4) as the exponential map $\operatorname{Exp} : T_\mathbb{U}\mathbb{G} \to \mathbb{G}$ and noting that, on a Riemannian manifold, the exponential map is a retraction, and that the product of exponential maps is again an exponential map Absil et al. (2009). For our sequence of *gradient-related* tangent vectors, we use the negative gradient, which is clearly always gradient-related. The gradient on the product manifold is the cartesian product of the gradients on each constituent manifold, i.e., $\boldsymbol{\nabla}(f) = (\boldsymbol{\nabla}(f_1), \ldots, \boldsymbol{\nabla}(f_n))$. Moreover, the inner product on the tangent space is the sum of the inner products on the constituent tangent spaces. Therefore, if $\{\boldsymbol{\Delta}_{i,t}\}$, $\boldsymbol{\Delta}_{i,t} \in T_{\mathbf{U}_t}\mathcal{M}_i$ is gradient-related for each $\mathcal{M}_i$, then $\{(\boldsymbol{\Delta}_{1,t}, \ldots, \boldsymbol{\Delta}_{n,t})\}$ is gradient-related on the product manifold. Furthermore, setting $\mathbb{U}_{t+1} = R_{\mathbb{U}_t}(\eta_t \boldsymbol{\Delta}_t)$ satisfies the bound in (7) with $c = 1$. Thus Theorem 2.1 follows as consequence of Theorem 4.3.1 and Corollary 4.3.2 in Absil et al. (2009). We refer the reader to Chapters 3 and 4 of Absil et al. (2009) for a review of the topics and definitions involved in this proof. $\square$

## 3 RELATED WORK

HRMC can be seen as the simultaneous generalization of low-rank matrix completion (LRMC) and subspace clustering (SC). In LRMC, one aims to recover the missing entries of a data matrix

whose columns lie in a single subspace (hence low-rank) Recht (2011a). HRMC is the generalization of LRMC to the case where the data columns lie in a union of subspaces Balzano et al. (2012a). Similarly, in SC the goal is to cluster a

collection of data columns lying in a union of subspaces Vidal (2011). HRMC is the generalization of SC to the case where data is incomplete (see Figure 2). Both LRMC and SC have been widely studied in the last years, resulting in a multitude of algorithms and guarantees for a wide variety of settings, covering diverse noise distributions McRae & Davenport (2021); Xia & Yuan (2021); Wang & Xu (2016), data distributions Fan et al. (2021); Qu & Xu (2015), privacy constraints Wang et al. (2015), outliers Huang

| | Number of Subspaces | |
| | 1 | **K** |
| Full-Data | PCA | SC |
| **Missing Data** | LRMC | **HRMC (This paper)** |

Figure 2: HRMC is a generalization of principal component analysis, LRMC, and SC.

et al. (2021); Peng et al. (2017), coherence assumptions Chen et al. (2014); Chen (2015), affinity learning Li et al. (2017); Tang et al. (2018), sampling schemes Balzano et al. (2010a); Pimentel-Alarcón & Nowak (2015), and more Vidal et al. (2005); Ma et al. (2008); Tseng (2000); Agarwal & Mustafa (2004); Tipping & Bishop (1999); Derksen et al. (2007). Consequently, for HRMC, if one could complete the data, one could then use a SC algorithm to identify the clusters and the underlying union. Similarly, if one could cluster the incomplete data, one could then use a LRMC algorithm on each cluster to fill the missing entries and identify each subspace. The challenge in HRMC is to simultaneously do both: cluster and complete.

**HRMC vs LRMC.** Given a HRMC problem, if the number of underlying subspaces, say $K$, and the maximum of their dimension, say $r$, are low, one could be tempted to cast HRMC as a LRMC problem. In such case, the single subspace containing all the columns of $\mathbf{X}$ would have dimension no larger than $r' := r \cdot K$. This would, however, completely ignore the union structure present in the data, and therefore require more observed entries in order to complete $\mathbf{X}$. We can see this by noting that each column must have more observed entries than the subspace containing it Pimentel-Alarcon & Nowak (2016). This means that even in the fortunate case where $r'$ is low enough, using LRMC would require $K$ times more observations than HRMC. This is especially prohibitive in applications such as Metagenomics or Drug Discovery, where data is extremely sparse and costly to acquire. In general, $r'$ may be too large to even allow the use of LRMC.

**HRMC vs SC.** Similarly, given a HRMC problem, a natural approach would be to fill-in the missing entries *naively* (with zeros, means, or LRMC) prior to clustering with any suitable full-data method. There exists a vast repertoire of SC theory and algorithms that guarantee perfect clustering under reasonable assumptions (e.g., sufficient sampling and subspace separation) Vidal et al. (2005); Ma et al. (2008); Tseng (2000); Agarwal & Mustafa (2004); Tipping & Bishop (1999); Derksen et al. (2007). Unfortunately, this approach may work if data is missing at a rate inversely proportional to the dimension of the subspaces Tsakiris & Vidal (2018), but fails with moderate volumes of missing data, as data filled naively no longer lies in a union of subspaces Elhamifar (2016).

**Tailored HRMC algorithms.** Algorithms specifically designed to solve the HRMC problem can be further divided in the following subgroups: (1) *neighborhood* methods that cluster points according to their overlapping coordinates Eriksson et al. (2012), (2) *alternating* methods, like EM Pimentel et al. (2014), $k$-subspaces Balzano et al. (2012b), group-lasso Pimentel-Alarcón et al. (2016); Saha et al. (2013), S$^3$LR Li & Vidal (2016), or MCOS Li et al. (2021) (3) *liftings*, which exploit the second-order algebraic structure of unions of subspaces Vidal et al. (2005; 2008); Ongie et al. (2017); Fan & Chow (2018); Fan & Udell (2019), and (4) *integer programming* Soni et al.. Neighborhood methods require either abundant observations or a super-polynomial number of samples (to produce enough overlaps). Liftings require squaring the dimension of an already high-dimensional problem, which severely limits their applicability. Integer programming approaches are similarly restricted to small data. To summarize, while much research has been devoted to HRMC, current algorithms have shortcomings, and little is known regarding their theoretical guarantees.

**Our work in context.** Among the methods discussed above, the approach of this paper is perhaps closer in principle to Mishra et al. (2019), which uses a similar Grassmannian optimization model to study the single-subspace problem of LRMC. This paper generalizes these ideas to the much harder multiple-subspace problem of HRMC, while maintaining the local convergence guarantees of Proposition 5.1 in Mishra et al. (2019). The main difference between Mishra et al. (2019) and our

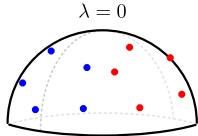 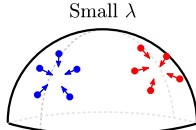 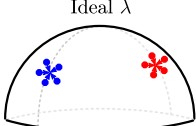 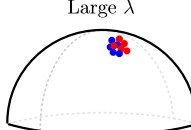

Figure 3: $\lambda \geq 0$ in (1) regulates how clusters fuse together. If $\lambda = 0$, each point is assigned to a subspace that exactly contains it (overfitting). The larger $\lambda$, the more we penalize subspaces being apart, which results in subspaces getting closer to form fewer clusters. The extreme case $\lambda = \infty$ is the special case of PCA and LRMC, where only one subspace is allowed to explain all data.

formulation is that the former only considers a predefined subset of geodesic distances (see equations (17)-(19) in Mishra et al. (2019)), which determine the Grassmannian points that must be matched. In Mishra et al. (2019), these subsets of geodesics can be chosen somewhat arbitrarily, because in LRMC all points belong to the same subspace. A so-called *gossip* protocol is therefore suitable in the easier problem of LRMC. In contrast, HRMC requires that only certain subsets of the Grassmannian be matched (the points corresponding to the unknown clustering to be learnt). Without knowing a priori the correct clusters, one cannot utilize the *gossip* method and must therefore use *all* pairwise geodesics so as to not introduce bias.

**Shortening the gap towards the fundamental sampling limit.** Theoretically, the sampling limit of HRMC is identical to that of LRMC except for a negligible *checksum* constant Pimentel-Alarcon & Nowak (2016). Intuitively, this means that HRMC should tolerate virtually the same amount of missing data as LRMC. However, in practice this is not the case. Contrary to LRMC algorithms, all HRMC algorithms fall short of this limit. An important open question that remains to be answered is whether it is possible to achieve the information-theoretic sampling limit with a polynomial-time algorithm, or if there is a fundamental statistical-computational trade-off that precludes such possibility. This paper shortens the gap towards the theoretical sampling limit (see Figures 4-5), thus shedding some light on a fundamental open question.

## 4 FUSION IN PRACTICE

**Clustering, completion, and subspace inference.** Recall that the solution to (1) provides an estimator $\mathbb{U}_i$ of $\mathbb{U}_i^\star$, the true subspace from which $\mathbf{x}_i$ is drawn. After solving (1), one may form the matrix $\mathbf{D}$ whose $(i, j)^{th}$ entry is given by $d_g(\mathbf{U}_i, \mathbf{U}_j)$ and use it as input to any distance-based clustering method, such as k-means Bottou & Bengio (1995); Elkan (2003); Sculley (2010), spectral clustering Tong et al. (2020); Veenstra et al. (2016); Von Luxburg (2007); Wen (2020), or DBSCAN Ester et al. (1996); Hou et al. (2016); Schubert et al. (2017). While prior knowledge of the number of subspaces K may be required for some clustering methods (e.g., k-means, or spectral clustering), it is not required at all to solve (1). Hence, by choosing a clustering method that doesn't require knowing K (e.g., DBSCAN), our approach can be applied to situations where K is unknown. After clustering, one can agglomerate all the data points corresponding to the $k^{th}$ cluster in the same matrix $\hat{\mathbf{X}}_k^\Omega$, and run any low-rank matrix completion (LRMC) algorithm (e.g., Balzano et al. (2010b); Bauch et al. (2021); Candes & Tao (2010); Candès & Recht (2009); Candès et al. (2011); Keshavan et al. (2010); Recht (2011b); Vidal & Favaro (2014)) to estimate its completion $\hat{\mathbf{X}}_k$. Finally, one can run principal component analysis (PCA) Minka (2000); Wold et al. (1987) on $\hat{\mathbf{X}}_k$ to recover an estimate basis $\hat{\mathbf{U}}_k$ of the $k^{th}$ underlying subspace $\mathbb{U}_k^\star$.

**Penalty parameter and model selection.** Intuitively, the chordal term in (1) forces each subspace to be close to its assigned data point, and the geodesic term forces subspaces from different data points to be close to one another. The tradeoff between these two quantities is determined by the penalty parameter $\lambda \geq 0$. If $\lambda = 0$, then the geodesic term is ignored and there is a trivial solution where each subspace exactly contains its assigned data point (thus attaining the minimum, zero, for the chordal distance). If $\lambda > 0$, the geodesic term forces subspaces from different data points to get closer, even if they no longer contain exactly their assigned data points. As $\lambda$ grows, subspaces get closer and closer (see Figure 3). The extreme case ($\lambda = \infty$) forces all subspaces to fuse into one (to attain zero in the second term), allowing only one subspace to explain all data, which is the equivalent

of PCA in the complete-data case, and LRMC if data is missing. In other words, PCA and LRMC are the special cases of our formulation with $\lambda = \infty$.

Ultimately, the effect of $\lambda$ will be reflected in the distance matrix $\mathbf{D}$, which in turn determines the number of clusters. The smaller $\lambda$, the more clusters, up to the extreme where each data point is in its own cluster. Conversely, the larger $\lambda$, the fewer clusters, up to the extreme point where all points are clustered together. The more subspaces, the more accuracy, but the more degrees of freedom (overfitting). To determine the best $\lambda$, one can compute a goodness of fit test, like the minimum effective dimension Huang et al. (2004); Vidal et al. (2005), that quantifies the tradeoff between accuracy and degrees of freedom. Similarly, we can iteratively increase r in (1) to find all the data points that lie in 1-dimensional subspaces, then all the data points that lie in 2-dimensional subspaces, and so on (pruning the data at each iteration). This will result in an estimate of the number of subspaces K, and their dimensions.

**Initialization.** In our implementation we initialize (1) with a solution to the problem when $\lambda = 0$, i.e., when each subspace perfectly contains the observed entries of its assigned data point. To this end, for each i we first construct an m × r matrix whose first column is equal to $\mathbf{x}_i^\Omega$ in its observed entries, and whose remaining entries are filled with standard normal entries, known to produce incoherent and uniformly distributed subspaces Eriksson et al. (2012). This matrix is then orthonormalized to produce the initial estimate $\mathbf{U}_i$, which, by construction, contains $\mathbf{x}_i^\Omega$, thus producing $d_c(\mathbf{x}_i^\Omega, \mathbf{U}_i) = 0$.

**Computational complexity.** We point out that the main caveat of our approach is its quadratic complexity in the number of samples. Fortunately, subspace clustering allows a simple approach to sketching both samples and features Traganitis & Giannakis (2017). That is, one may solve (1) with a subset of $n' \leq n$ columns, and a subset of $m' \leq m$ rows (e.g., those with most observations), resulting in an improved complexity, quadratic in $n'$ as opposed to n. With the solution of (1), one can use a clustering method, a LRMC algorithm, and PCA, as described above, to produce subspace estimates $\hat{\mathbf{U}}_1, \ldots, \hat{\mathbf{U}}_{K'}$, with $K' \leq K$. Each of the remaining $n - n'$ incomplete data points $\mathbf{x}_i^\Omega$ that were not used to solve (1) and that have more than r observations (a fundamental requirement of subspace clustering Pimentel-Alarcon & Nowak (2016)) can be trivially assigned to the subspace estimate producing the largest projection coefficient $\boldsymbol{\theta}_i^k = (\hat{\mathbf{U}}_k^{\Omega\mathsf{T}}\hat{\mathbf{U}}_k^\Omega)^{-1}\mathbf{x}_i^\Omega$, where $\hat{\mathbf{U}}_k^\Omega \in \mathbb{R}^{|\Omega_i| \times r}$ denotes the restriction of $\hat{\mathbf{U}}_k$ to the observed rows of $\mathbf{x}_i^\Omega$ (notice that $\hat{\mathbf{U}}_k^{\Omega\mathsf{T}}\hat{\mathbf{U}}_k^\Omega$ is invertible for almost every rank-r $\hat{\mathbf{U}}_k$ whenever $|\Omega_i| > r$ Pimentel-Alarcon & Nowak (2016)). If $\mathbf{x}_i^\Omega$ is assigned to $\hat{\mathbf{U}}_k$, its completion can be trivially estimated as $\hat{\mathbf{x}}_i = \hat{\mathbf{U}}_k\boldsymbol{\theta}_i^k$. All the data points $\mathbf{x}_i^\Omega$ that are too far from all of the subspace estimates (equivalently, the data points whose coefficients are smaller than a pre-determined parameter) can be used to solve (1) again for a refined clustering.

## 5 EXPERIMENTS

In this section we present a series of experiments on real and synthetic data, in particular the Hopkins155 dataset Tron & Vidal (2007), and the Smartphone dataset for Human Activity Recognition in Ambient Assisted Living (AAL) Anguita et al. (2013a). Rather than establishing a new state-of-the-art, these experiments have the intention to serve as proof of concept, showing the potential of our approach, which in this first introduction and basic formulation performs comparable to prominent methods Yang et al. (2015). In our experiments we initialize (1) as described in Section 4, with r fixed, as is known a priori in both, the simulations, and the real datasets. We do not specify K, and we make no special adjustments to handle noise, as is not required by our approach. The attained solution to (1) is used as input to spectral clustering Tong et al. (2020); Veenstra et al. (2016); Von Luxburg (2007); Wen (2020) (though, as described in Section 4, other clustering algorithms, such as k-means Bottou & Bengio (1995); Elkan (2003); Sculley (2010) or DBSCAN Ester et al. (1996); Hou et al. (2016); Schubert et al. (2017) could be used). We measure accuracy in terms of clustering error, given by $\min_M \frac{1}{n} \sum_{i=1}^n \mathbb{1}_{\{M(\hat{\mathbf{y}}) \neq \mathbf{y}\}}$, where $\mathbb{1}$ denotes the indicator function, and $M$ is a function that maps the estimated cluster labels $\hat{\mathbf{y}} \in \{1, \ldots, \hat{K}\}^n$ assigned to $\mathbf{x}_1^\Omega, \ldots, \mathbf{x}_n^\Omega$, to the true labels $\mathbf{y} \in \{1, \ldots, K\}^n$. All trials outlined in this section are expected to be completed within a maximum time frame of two hours on a single-CPU server machine.

**Baseline comparisons.** A recent survey Lane et al. (2019b) shows that most state-of-the-art algorithms for HRMC (including MC+SSC Yang et al. (2015), EM Pimentel et al. (2014), GSSC Pimentel-Alarcón et al. (2016), $k$-subspaces Balzano et al. (2012b), and more) have similar perfor-

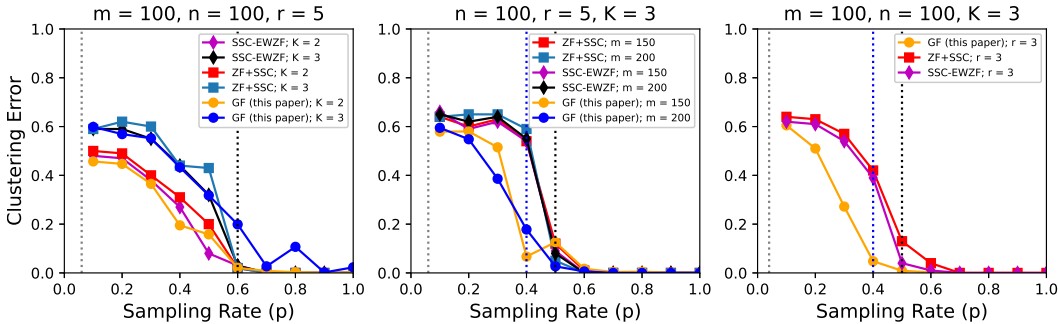

Figure 4: Clustering error (average over 10 trials) as a function of sampling rate for different synthetic settings. The left-most vertical line at $p^\star = (r+1)/\min(m,n)$ represents the information-theoretic sampling limit Pimentel-Alarcon & Nowak (2016). That is, HRMC is impossible for any $p < p^\star$, and is theoretically possible for any $p \geq p^\star$ (for example, with a brute-force combinatorial algorithm). The right-most vertical line indicates the limit of the state-of-the-art. The center vertical line indicates the sampling limit of our approach, which shortens the gap towards the theoretical limit.

mance, with varying winners on specific scenarios depending on subspace vs ambient dimension gap, fraction of missing data, and number of subspaces. Based on this recent survey Lane et al. (2019b), and others Pimentel-Alarcón et al. (2016), we chose ZF+SSC and SSC-EWZF as baselines, which has been seen in Lane et al. (2019b) to have nearly identical performance as MC+SSC, EM, GSSC, $k$-subspaces and k-GROUSE Balzano et al. (2012b) in the scenarios discussed in our paper.

**Synthetic data.** In all our simulations we first generate K matrices $\mathbf{U}_k^\star \in \mathbb{R}^{m \times r}$ with i.i.d. $\mathcal{N}(0,1)$ entries, to use as bases of the *true* subspaces. For each $k$ we generate a matrix $\mathbf{\Theta}_k^\star \in \mathbb{R}^{r \times n_k}$, also with i.i.d. $\mathcal{N}(0,1)$ entries, to use as coefficients of the columns in the $k^{\text{th}}$ subspace. We then form $\mathbf{X}$ as the concatenation $[\mathbf{U}_1^\star \mathbf{\Theta}_1^\star, \ \mathbf{U}_2^\star \mathbf{\Theta}_2^\star, \ \ldots, \ \mathbf{U}_K^\star \mathbf{\Theta}_K^\star]$. To induce missing data we sample each entry independently with probability $p$. Figure 4 shows the clustering results as a function of the sampling rate for a variety of settings, tuning the parameter $\lambda$ manually. Notice that, even with this first formulation, we perform comparable to existing methods, and even better in some cases, especially in low-sampling regimes.

**Object tracking in Hopkins 155.** This dataset contains 155 videos of K = 2 or K = 3 moving objects, such as checkerboards, vehicles, and pedestrians. In each video, a collection of $n$ mark points are tracked through all frames. The locations over time of the $i^{\text{th}}$ point are stacked to produce $\mathbf{x}_i \in \mathbb{R}^m$, so that the points corresponding to the same object lie near a low-dimensional subspace Tomasi & Kanade (1992); Kanatani (2001) ($r$ varies from video to video, from 1 to 3). In all cases we fixed the penalty parameter $\lambda$ to 1. To induce missing data (e.g., produced by occlusions) we sample each entry independently with probability $p$. Figure 5 shows the clustering results.

**Human activity recognition in Smartphone AAL Dataset.** This dataset contains $n = 5744$ instances, each with $m = 561$ features related to pre-processed accelerometer and gyroscope time series and summary statistics Anguita et al. (2013b), related to K = 2 activities: walking, and other movements, each approximated by a subspace of dimension $r = 4$. Recall that the complexity of (1) is quadratic in $n$, so if solved directly, this dataset that would produce an unmanageable computational complexity. However, using the sketching techniques described in Section 4, it can be solved quite efficiently. In particular, we only used $m' = 158$ features (related to the accelerometer's and gyroscope's minimum, maximum, standard deviation, and mean parameters over time), and $n' = 100$ samples selected at random, evenly distributed among classes. In all cases we fixed the penalty parameter $\lambda$ to $10^{-5}$. The results are summarized in Figure 5. Notice that our approach outperforms existing methods in the low-sampling regime.

Lastly, we note the disparity in performance between our model and existing algorithms when the missing data rate is low. The main motivation for our approach is incomplete data. While our formulation can certainly be used with full data, we acknowledge that it would be an over-kill, and consequently suboptimal in that scenario, which has been extensively studied. Hence, it is not

surprising that methods tailored for full-data outperform ours in such setting. However, no full-data method outperforms ours when data is missing.

## 6 FUTURE DIRECTIONS AND CHALLENGES

The main formulation presented in this paper is a non-convex optimization that relies on the simultaneous interactions of many terms. A proper analysis of the model is therefore challenging. One difficulty confronted is the complex geometry of the zero-set of the chordal distance term, or more precisely, the intersections of many of these zero-sets, one for each column of data. While intuition suggests that the model is encouraged by the geodesic term to find regions of "dense" intersection, and therefore build consensus, a more precise formulation of this intuition has evaded us. This is further highlighted in Figures 4 and 5 by the fact that the performance of our model decreases as K grows, indicating that it is not currently understood how the combination of the chordal and geodesic terms encourage consensus amongst so many cross-cluster terms. It is our belief that a well-designed weighted version of (1), such as

$$\min_{\mathbf{U}_1,\dots,\mathbf{U}_n \in \mathbb{S}^{m\times r}} \quad \sum_{i=1}^{n} d_c^2(\mathbf{x}_i^\Omega, \mathbf{U}_i) \; + \; \frac{\lambda}{2} \sum_{i,j=1}^{n} w_{ij} d_g^2(\mathbf{U}_i, \mathbf{U}_j), \tag{5}$$

where the weights $w_{ij} \geq 0$ quantify how much attention is given to each penalty, is key to unlocking better performance and understanding of the model. Our immediate future work will focus on investigating options for these weights, such as inverse distance functions, or k-nearest neighbors, known to dramatically improve the performance, computational complexity, and tolerance to K of fusion formulations in Euclidean space Antoniadis & Fan (2001); Chi & Lange (2015); Tan & Witten (2015); Yuan & Lin (2006).

## 7 CONCLUSIONS

This paper presents a new paradigm for clustering incomplete datapoints using subspaces as proxies to leverage the geometry of the Grassmannian. This new perspective enables clustering and completion of data in a union of subspaces. This work should be understood as the first introduction to the idea of fusion penalties in the Grassmann manifold, for the problem of *high-rank matrix completion*. Rather than establishing our approach as the state-of-the-art, our experiments have the intention to serve as proof of concept, showing that there is potential in our approach, in the hopes to ignite future work on several directions, such as the study of weighted versions described in (5), the choice of penalty parameters, and variants robust to outliers.

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

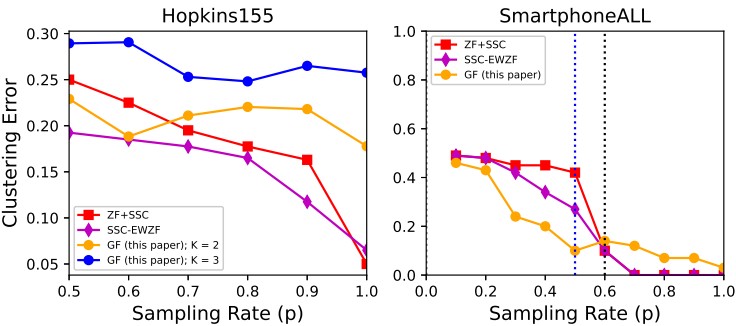

Figure 5: Clustering error as a function of sampling rate for real datasets. **Left:** average over 120 videos with $K = 2$ objects, and 35 videos with $K = 3$ objects. **Right:** average over 20 trials. The left-most vertical line at $p^\star = (r+1)/\min(m, n)$ represents the information-theoretic sampling limit Pimentel-Alarcon & Nowak (2016). That is, HRMC is impossible for any $p < p^\star$, and is theoretically possible for any $p \geq p^\star$ (for example, with a brute-force combinatorial algorithm). The right-most vertical line indicates the limit of the state-of-the-art. The center vertical line indicates the sampling limit of our approach, which shortens the gap towards the theoretical limit.

## A    EXPLICIT HRMC EXAMPLE

This section presents an explicit example of the HRMC setup and goals. Suppose we observe the following data matrix, where the unobserved entries are marked with $*$:

$$\begin{bmatrix} 1 & -4 & 6 & 9 & 16 & * & 8 & * & * \\ * & 5 & 4 & * & 16 & 5 & * & * & * \\ * & 7 & 6 & -12 & 2 & 18 & -1 & * & * \\ 2 & * & 5 & * & * & * & 1 & 0 & -1 \end{bmatrix},$$

The goal of HRMC is to $(i)$ obtain the following (ground truth) completion:

$$\mathbf{X} = \begin{bmatrix} 1 & -4 & 6 & 9 & 16 & -1 & 8 & -7 & 3 \\ 1 & 5 & 4 & 14 & 16 & 5 & -7 & -18 & 10 \\ 8 & 7 & 6 & -12 & 2 & 18 & -1 & 28 & -18 \\ 2 & 1 & 5 & 4 & 11 & 4 & 1 & 0 & -1 \end{bmatrix}.$$

We should also $(ii)$ cluster the columns of $\mathbf{X}$ into two groups, $\{\mathbf{x}_1, \mathbf{x}_2, \mathbf{x}_6, \mathbf{x}_7\}$ and $\{\mathbf{x}_3, \mathbf{x}_4, \mathbf{x}_5, \mathbf{x}_8, \mathbf{x}_9\}$, and $(iii)$ obtain bases for two 2-dimensional subspaces (given by any subset of linearly independent columns from each group).

## B    GRASSMANN AND STIEFEL MANIFOLDS

In the interest of self-containment, this section summarizes the main fundamental concepts required to understand the tools used in this paper. A detailed exposition on these topics is presented in Edelman et al. (1998). The primary mathematical object involved in this work is the Grassmannian $\mathbb{G}(m, r)$. This is a smooth compact manifold of dimension $r(m - r)$. To describe this, it is necessary to define precursor objects; the orthogonal group $O(m)$ and the Stiefel manifold $\mathbb{S}(m, r)$. The objects of interest are thus:

1. The orthogonal group $O(m)$ consisting of $m \times m$ orthogonal matrices;

2. The Stiefel manifold $\mathbb{S}(m, r)$ consisting of $m \times r$ orthonormal matrices;

3. The Grassmann manifold $\mathbb{G}(m, r)$ obtained by identifying those matrices in $\mathbb{S}(m, r)$ whose columns span the same subspace (a quotient manifold).

In this setting, the Stiefel manifold is defined as a quotient space of the orthogonal group. Here, two orthogonal matrices $\mathbf{U}$ and $\mathbf{V}$ are identified if their first $r$ columns are identical or, equivalently, if $\mathbf{U} = \left( \begin{smallmatrix} \mathbf{I} & 0 \\ 0 & \mathbf{Q} \end{smallmatrix} \right) \mathbf{V}$, where $\mathbf{Q}$ is an orthogonal $(m - r) \times (m - r)$ block. Therefore $\mathbb{S}(m, r) = O(m)/O(m - r)$. Going further, the Grassmannian is defined a quotient space of the Stiefel manifold where two Stiefel elements are identified if their columns span the same $r$-dimensional subspace. Therefore $\mathbb{G}(m, r) = \mathbb{S}(m, r)/O(r)$.

Given the above, it is clear that we may describe elements of the Stiefel and Grassmann manifolds using concrete representatives that can be stored on a computer. A point on the Stiefel manifold may be stored as an $m \times r$ orthonormal matrix. A point on the Grassmann manifold, however, being a linear subspace, does not have a unique representative and can be stored as an arbitrary $m \times r$ orthonormal matrix so long as it spans the correct subspace.

## C    PRINCIPAL ANGLES AND SINGULAR VALUES

Recall the notion of principal angles between subspaces: let $\mathbf{U} \in \mathbb{S}(m, p)$ and $\mathbf{V} \in \mathbb{S}(m, q)$ be orthonormal bases for two arbitrary subspaces of $\mathbb{R}^m$. Assume, without loss of generality, that $1 \leq p \leq q \leq m$. The principal angles between $\mathrm{span}(\mathbf{U})$ and $\mathrm{span}(\mathbf{V})$ are defined via the following construction. Let $\mathbf{u}_1 \in \mathrm{span}(\mathbf{U})$ and $\mathbf{v}_1 \in \mathrm{span}(\mathbf{V})$ be unit vectors such that $|\mathbf{u}_1^\mathsf{T} \mathbf{v}_1|$ is maximal. Inductively, let $\mathbf{u}_k \in \mathrm{span}(\mathbf{U})$ and $\mathbf{v}_k \in \mathrm{span}(\mathbf{V})$ be unit vectors such that $\mathbf{u}_k^\mathsf{T} \mathbf{u}_j = 0$ and $\mathbf{v}_k^\mathsf{T} \mathbf{v}_j = 0$ for all $1 \leq j < k$ and $|\mathbf{u}_k^\mathsf{T} \mathbf{v}_k|$ is maximal. The principal angles are defined as $\alpha_k = \arccos \mathbf{u}_k^\mathsf{T} \mathbf{v}_k$ for all $k = 1, 2, \ldots, p$.

This constructive definition is too cumbersome to use in practice. We opt for the following alternative computation via the singular value decomposition. Let $\mathbf{u}_1, \ldots, \mathbf{u}_p$ and $\mathbf{v}_1, \ldots, \mathbf{v}_q$ be the columns of

$\mathbf{U}$ and $\mathbf{V}$ respectively. Compute the singular value decomposition $\mathbf{U}^\mathsf{T}\mathbf{V} = \bar{\mathbf{U}}\mathrm{diag}[\ldots, \sigma_i, \ldots]\bar{\mathbf{V}}^\mathsf{T}$. Set $\mathbf{U}' = \mathbf{U}\bar{\mathbf{U}}$ and $\mathbf{V}' = \mathbf{V}\bar{\mathbf{V}}$ and denote their columns by $\mathbf{u}'_i$ and $\mathbf{v}'_j$ respectively. Observe that $\mathrm{span}(\mathbf{U}) = \mathrm{span}(\bar{\mathbf{U}})$ and $\mathrm{span}(\mathbf{V}) = \mathrm{span}(\bar{\mathbf{V}})$ and furthermore that $\mathbf{U}'^\mathsf{T}\mathbf{V}' = \mathrm{diag}[\ldots, \sigma_i, \ldots]$, that is

$$\mathbf{u}'^\mathsf{T}_i \mathbf{v}'_j = \begin{cases} \sigma_i & i = j \\ 0 & i \neq j. \end{cases}$$

The vectors $\mathbf{u}'_i$ and $\mathbf{v}'_j$ correspond to those in the constructive definition. We therefore have that the $i$-th principal angle $\alpha_i$ relates to the $i$-th singular value via $\sigma_i = \cos\alpha_i$.

## D  CHORDAL DISTANCE

The chordal distance between points on the Grassmannian $\mathbb{G}(m, r)$, introduced and studied in Conway et al. (1996), is defined via

$$\sqrt{\sum_{i=1}^{r} \sin^2\alpha_i},$$

where the $\alpha_i$ are the principal angles between the points described above. The authors of Wei Dai & Milenkovic (2012) introduce a notion of distance between a partially observed vector $\mathbf{x}^\Omega \in \mathbb{R}^m$ and a subspace $\mathbf{U} \in \mathbb{G}(m, r)$ via a formulation closely related to the chordal distance, which they give the same name.

Let $\mathbf{X}^0$ denoted the orthonormal matrix spanning all possible completions of $\mathbf{x}^\Omega$: If $\mathbf{x}^\Omega = \mathbf{0}$, then $\mathbf{X}^0 = \mathbf{I}$, the identity matrix. Otherwise, $\mathbf{X}^0$ is the $\mathrm{m} \times (\mathrm{m} - |\Omega| + 1)$ matrix formed with $\mathbf{x}^\Omega$ normalized and filled with zeros in the unobserved rows, concatenated with the $(\mathrm{m} - |\Omega|)$ canonical vectors indicating the unobserved rows of $\mathbf{x}^\Omega$. Let $\sigma_1(\mathbf{X}^{0T}\mathbf{U})$ denote the largest singular value of $\mathbf{X}^{0T}\mathbf{U}$. Then

$$d_c(\mathbf{x}^\Omega, \mathbf{U}) = \sin\alpha_1 = \sqrt{1 - \sigma_1^2(\mathbf{X}^{0T}\mathbf{U})}.$$

This metric is studied in Wei Dai & Milenkovic (2012). Of particular importance is the following fact stated as Theorem 2 in Wei Dai & Milenkovic (2012): the preimage of 0 under $d_c^2(\mathbf{x}^\Omega, \cdot)$ is the closure of the preimage of 0 under $f_F(\mathbf{x}^\Omega, \cdot)$, where

$$f_F(\mathbf{x}^\Omega, \mathbf{U}) = \min_{w \in \mathbb{R}^r} \|x^\Omega - \mathcal{P}_\Omega(\mathbf{U}w)\|_F^2,$$

and $\mathcal{P}_\Omega$ denotes projection onto the entries indexed by $\Omega$. That is, $f_F$ is the Frobenious norm, which is often used to search for subspaces $\mathbf{U}$ consistent with data. The Frobenius norm may not be continuous, whereas the chordal distance is continuous and differentiable.

## E  GEODESIC DISTANCE ON THE GRASSMANNIAN

The geodesic distance $d_g(\mathbf{U}_i, \mathbf{U}_j)$ is derived from the intrinsic geometry of the Grassmann manifold and depends on the metric which defines the manifold structure. Let $\gamma : [a, b] \to \mathcal{M}$ be a curve on a general Riemannian manifold $(\mathcal{M}, g)$ with metric $g$. Then the length of $\gamma$ is defined as John (2018)

$$L(\gamma) = \int_a^b \sqrt{g(\dot{\gamma}(t), \dot{\gamma}(t))}dt.$$

The canonical metric for the Grassmann manifold coincides with the Euclidean metric inherited from $O(m)$: $g_c(\dot{\mathbf{U}}_i, \dot{\mathbf{U}}_j) = g_e(\dot{\mathbf{U}}_i, \dot{\mathbf{U}}_j) = \mathrm{tr}(\dot{\mathbf{U}}_i^\mathsf{T}\dot{\mathbf{U}}_j)$ Edelman et al. (1998). To compute the geodesic distance, we therefore require knowledge of the geodesic segment connecting $\mathbf{U}_i$ and $\mathbf{U}_j$ with respect to the metric $g_c$. This is described in Lemma 1 of Wei Dai & Milenkovic (2012): Let $\mathbf{V}_i\boldsymbol{\Sigma}\mathbf{V}_j^\mathsf{T}$ be the singular value decomposition of the matrix $\mathbf{U}_i^\mathsf{T}\mathbf{U}_j$, and denote the $\ell$-th singular value by $\sigma_\ell = \cos\alpha_\ell$. Set $\bar{\mathbf{U}}_i = \mathbf{U}_i\mathbf{V}_i$ and $\bar{\mathbf{U}}_j = \mathbf{U}_j\mathbf{V}_j$ and note that $\bar{\mathbf{U}}_i^\mathsf{T}\bar{\mathbf{U}}_j = \boldsymbol{\Sigma}$. Then the geodesic with respect to $g_c$ from $\mathbf{U}_i$ to $\mathbf{U}_j$ is given by $\mathbf{U}(t), 0 \leq t \leq 1$, where the path $\mathbf{U}(t)$ is given by

$$[\bar{\mathbf{U}}_i, \mathbf{G}] \begin{bmatrix} \mathrm{diag}\,([\ldots, \cos\alpha_\ell t, \ldots]) \\ \mathrm{diag}\,([\ldots, \sin\alpha_\ell t, \ldots]) \end{bmatrix} \mathbf{V}_i^\mathsf{T},$$

where the columns of $\mathbf{G} = [\dots, \boldsymbol{g}_\ell, \dots] \in \mathbb{S}(r, m)$ are defined as

$$\boldsymbol{g}_\ell = \begin{cases} \frac{\bar{\mathbf{U}}_{2,:\ell} - \sigma_i \bar{\mathbf{U}}_{1,:\ell}}{\|\bar{\mathbf{U}}_{2,:\ell} - \sigma_\ell \bar{\mathbf{U}}_{1,:\ell}\|} & \text{if } \lambda_\ell \neq 1 \\ \mathbf{0} & \text{if } \lambda_\ell = 1. \end{cases}$$

Here, the subscript $: \ell$ denotes the $\ell$-th column of the corresponding matrix.

We therefore have

$$\dot{\mathbf{U}}(t) = [\bar{\mathbf{U}}_\mathrm{i}, \mathbf{G}] \left[ \begin{array}{c} \mathrm{diag}\left([\dots, -\alpha_\ell \sin \alpha_\ell t, \dots]\right) \\ \mathrm{diag}\left([\dots, \quad \alpha_\ell \cos \alpha_\ell t, \dots]\right) \end{array} \right] \mathbf{V}_\mathrm{i}^\mathsf{T}.$$

Denote $\mathbf{S} = \mathrm{diag}\left([\dots, -\alpha_\ell \sin \alpha_\ell t, \dots]\right)$ and $\mathbf{C} = \mathrm{diag}\left([\dots, \alpha_\ell \cos \alpha_\ell t, \dots]\right)$. Then

$$\begin{aligned} \dot{\mathbf{U}}^\mathsf{T} \dot{\mathbf{U}} &= \mathbf{V}_\mathrm{i}[\mathbf{S}, \mathbf{C}] \left[ \begin{array}{c} \bar{\mathbf{U}}_\mathrm{i}^\mathsf{T} \\ \mathbf{G}^\mathsf{T} \end{array} \right] [\bar{\mathbf{U}}_\mathrm{i}, \mathbf{G}] \left[ \begin{array}{c} \mathbf{S} \\ \mathbf{C} \end{array} \right] \mathbf{V}_\mathrm{i}^\mathsf{T} \\ &= \mathbf{V}_\mathrm{i}[\mathbf{S}, \mathbf{C}] \left[ \begin{array}{cc} \bar{\mathbf{U}}_\mathrm{i}^\mathsf{T} \bar{\mathbf{U}}_\mathrm{i} & \bar{\mathbf{U}}_\mathrm{i}^\mathsf{T} \mathbf{G} \\ \mathbf{G}^\mathsf{T} \bar{\mathbf{U}}_\mathrm{i} & \mathbf{G}^\mathsf{T} \mathbf{G} \end{array} \right] \left[ \begin{array}{c} \mathbf{S} \\ \mathbf{C} \end{array} \right] \mathbf{V}_\mathrm{i}^\mathsf{T} \\ &= \mathbf{V}_\mathrm{i}(\mathbf{S}^2 + \mathbf{C}^2)\mathbf{V}_\mathrm{i}^\mathsf{T} \\ &= \mathbf{V}_\mathrm{i}\mathrm{diag}([\dots, \alpha_\ell^2, \dots])\mathbf{V}_\mathrm{i}^\mathsf{T}, \end{aligned}$$

where we use the fact that $\bar{\mathbf{U}}_\mathrm{i}, \mathbf{G} \in \mathbb{S}(m, r)$, and that $\bar{\mathbf{U}}_\mathrm{i}^\mathsf{T} \mathbf{G} = \mathbf{0}$ Wei Dai & Milenkovic (2012). Recall that $\mathrm{tr}(AB) = \mathrm{tr}(BA)$, hence $\mathrm{tr}(\dot{\mathbf{U}}^\mathsf{T} \dot{\mathbf{U}}) = \mathrm{tr}(\mathrm{diag}([\dots, \alpha_\ell^2, \dots])) = \sum_\ell \alpha_\ell^2$. We therefore have $L(\mathbf{U}(t)) = \int_0^1 \sqrt{\sum_i \alpha_\ell^2} dt = \sqrt{\sum_\ell \alpha_\ell^2}$. Recalling that $\sigma_\ell = \cos \alpha_\ell$, we finally have

$$d_g(\mathbf{U}_\mathrm{i}, \mathbf{U}_\mathrm{j}) \;=\; \sqrt{\sum_{\ell=1}^{\mathrm{r}} \mathrm{arccos}^2 \, \sigma_\ell(\mathbf{U}_\mathrm{i}^\mathsf{T} \mathbf{U}_\mathrm{j})}.$$

## F  GRADIENTS ON THE GRASSMANNIAN

In this section, we derive the expressions in (2) and (3) that govern the fusion steps of our formulation. For a function $F(\mathbf{U})$ defined on the Grassmannian, the graduate of $F$ at $\mathbf{U}$ is given by equation (2.70) in Edelman et al. (1998), which we record here:

$$\nabla F = F_\mathbf{U} - \mathbf{U}\mathbf{U}^\mathsf{T} F_\mathbf{U},$$

where $F_\mathbf{U}$ is the matrix whose entries are given by $[F_\mathbf{U}]_\mathrm{ij} = \frac{\partial F}{\partial \mathbf{U}_\mathrm{ij}}$.

**Chordal gradient.** To obtain the gradient of the chordal distance $d_c^2(\mathbf{x}_\mathrm{i}^\Omega, \mathbf{U}_\mathrm{i})$ presented in (2), consider the partial derivative with respect to the $(\mathrm{a}, \mathrm{b})^\mathrm{th}$ element of $\mathbf{U}_\mathrm{i}$:

$$\left[ \frac{\partial d_c^2(\mathbf{x}_\mathrm{i}^\Omega, \mathbf{U}_\mathrm{i})}{\partial \mathbf{U}_\mathrm{i}} \right]_\mathrm{ab} \;=\; \frac{\partial}{\partial [\mathbf{U}_\mathrm{i}]_\mathrm{ab}} d_c^2(\mathbf{x}_\mathrm{i}^\Omega, \mathbf{U}_\mathrm{i}) \;=\; -2\sigma_1(\mathbf{X}_\mathrm{i}^{0\mathsf{T}}\mathbf{U}_\mathrm{i}) \frac{\partial \sigma_1(\mathbf{X}_\mathrm{i}^{0\mathsf{T}}\mathbf{U}_\mathrm{i})}{\partial [\mathbf{U}_\mathrm{i}]_\mathrm{ab}}.$$

To obtain the partial derivative of the leading singular value $\sigma_1$, observe that $\mathbf{X}_\mathrm{i}^0 \mathbf{X}_\mathrm{i}^{0\mathsf{T}}\mathbf{U}_\mathrm{i}$ and $\mathbf{X}_\mathrm{i}^{0\mathsf{T}}\mathbf{U}_\mathrm{i}$ share singular values: if $\mathbf{X}_\mathrm{i}^{0\mathsf{T}}\mathbf{U}_\mathrm{i} = \mathbf{V}\boldsymbol{\Sigma}\mathbf{W}^\mathsf{T}$, then $\mathbf{X}_\mathrm{i}^0 \mathbf{X}_\mathrm{i}^{0\mathsf{T}}\mathbf{U}_\mathrm{i} = (\mathbf{X}_\mathrm{i}^0 \mathbf{V})\boldsymbol{\Sigma}\mathbf{W}^\mathsf{T}$ with $\mathbf{X}_\mathrm{i}^0 \mathbf{V} \in \mathbb{S}(m, r)$, so the result is a compact singular value decomposition. Recall that $\mathbf{v}_\mathrm{i}$ and $\mathbf{w}_\mathrm{i}$ denote the leading left and right singular vectors of $\mathbf{X}_\mathrm{i}^0 \mathbf{X}_\mathrm{i}^{0\mathsf{T}}\mathbf{U}_\mathrm{i}$. Since $\mathbf{v}_\mathrm{i}^\mathsf{T} \mathbf{X}_\mathrm{i}^0 \mathbf{X}_\mathrm{i}^{0\mathsf{T}}\mathbf{U}_\mathrm{i}\mathbf{w}_\mathrm{i} = \sigma_1$, we have that

$$\frac{\partial \sigma_1}{\partial [\mathbf{U}_\mathrm{i}]_\mathrm{ab}} \;=\; \frac{\partial \mathbf{v}_\mathrm{i}^\mathsf{T}}{\partial [\mathbf{U}_\mathrm{i}]_\mathrm{ab}} \mathbf{X}_\mathrm{i}^0 \mathbf{X}_\mathrm{i}^{0\mathsf{T}}\mathbf{U}_\mathrm{i}\mathbf{w}_\mathrm{i} \;+\; \mathbf{v}_\mathrm{i}^\mathsf{T}\mathbf{X}_\mathrm{i}^0 \mathbf{X}_\mathrm{i}^{0\mathsf{T}} \frac{\partial \mathbf{U}_\mathrm{i}}{\partial [\mathbf{U}_\mathrm{i}]_\mathrm{ab}} \mathbf{w}_\mathrm{i} \;+\; v_\mathrm{i}^\mathsf{T}\mathbf{X}_\mathrm{i}^0 \mathbf{X}_\mathrm{i}^{0\mathsf{T}}\mathbf{U}_\mathrm{i} \frac{\partial \mathbf{w}_\mathrm{i}}{\partial [\mathbf{U}_\mathrm{i}]_\mathrm{ab}}.$$

The first and third terms are zero, because $\mathbf{w}_\mathrm{i}$ is the leading right singular vector of $\mathbf{X}_\mathrm{i}^0 \mathbf{X}_\mathrm{i}^{0\mathsf{T}}\mathbf{U}_\mathrm{i}$, so $(\mathbf{X}_\mathrm{i}^0 \mathbf{X}_\mathrm{i}^{0\mathsf{T}}\mathbf{U}_\mathrm{i})\mathbf{w}_\mathrm{i} = \sigma_1\mathbf{v}_\mathrm{i}$, which implies

$$\frac{\partial \mathbf{v}_\mathrm{i}^\mathsf{T}}{\partial [\mathbf{U}_\mathrm{i}]_\mathrm{ab}}(\mathbf{X}_\mathrm{i}^0 \mathbf{X}_\mathrm{i}^{0\mathsf{T}}\mathbf{U}_\mathrm{i})\mathbf{w}_\mathrm{i} \;=\; \sigma_1 \frac{\partial \mathbf{v}_\mathrm{i}^\mathsf{T}}{\partial [\mathbf{U}_\mathrm{i}]_\mathrm{ab}}\mathbf{v}_\mathrm{i},$$

and because $\frac{\partial \mathbf{v}_i^\mathsf{T}}{\partial [\mathbf{U}_i]_{ab}} \mathbf{v}_i = 0$, as seen by differentiating both sides of $\mathbf{v}_i^\mathsf{T} \mathbf{v}_i = 1$ (and similarly for the third term). To compute the second term, note that $\mathbf{v}_i \in \mathrm{span}(\mathbf{X}_i^0)$ since it is a column of $\mathbf{X}_i^0 \mathbf{V}$, and the space spanned by $\mathbf{X}_i^0$ is invariant under multiplication by $\mathbf{V}$. Now, $(\mathbf{v}_i^\mathsf{T} \mathbf{X}_i^0 \mathbf{X}_i^{0\mathsf{T}})^\mathsf{T} = \mathbf{X}_i^0 \mathbf{X}_i^{0\mathsf{T}} \mathbf{v}_i = \mathbf{v}_i$, since $\mathbf{X}_i^0 \mathbf{X}_i^{0\mathsf{T}}$ acts on vectors as the projection onto $\mathrm{span}(\mathbf{X}_i^0)$. Hence $\mathbf{v}_i^\mathsf{T} \mathbf{X}_i^0 \mathbf{X}_i^{0\mathsf{T}} = \mathbf{v}_i^\mathsf{T}$. The second term then becomes $\mathbf{v}_i^\mathsf{T} \frac{\partial \mathbf{U}_i}{\partial [\mathbf{U}_i]_{ab}} \mathbf{w}_i = [\mathbf{v}_i]_a [\mathbf{w}_i]_b$. It follows that

$$\frac{\partial \sigma_1}{\partial [\mathbf{U}_i]_{ab}} = [\mathbf{v}_i]_a [\mathbf{w}_i]_b. \tag{6}$$

From this, we have $\boldsymbol{\nabla} d_c^2(\mathbf{x}_i^\Omega, \mathbf{U}_i) = -2(\mathbf{I} - \mathbf{U}_i \mathbf{U}_i^\mathsf{T}) \sigma_1 \mathbf{v}_i \mathbf{w}_i^\mathsf{T}$, where multiplication by $\mathbf{I} - \mathbf{U}_i \mathbf{U}_i^\mathsf{T}$ projects onto the tangent space of the Grassmannian at $\mathbf{U}_i$, as described before Absil et al. (2009); Edelman et al. (1998).

**Geodesic gradient.** For the gradient of the geodesic distance $d_g^2(\mathbf{U}_i, \mathbf{U}_j)$ in (3) let us use $\sigma_\ell$ as shorthand for $\sigma_\ell(\mathbf{U}_i^\mathsf{T} \mathbf{U}_j)$, and recall that $\mathbf{v}_{ij}^\ell$ and $\mathbf{w}_{ij}^\ell$ denote the $\ell^{\mathrm{th}}$ left and right singular vectors of $\mathbf{U}_j \mathbf{U}_j^\mathsf{T} \mathbf{U}_i$. Then the partial derivative with respect to the $(a, b)^{th}$ element of $\mathbf{U}_i$ is

$$\left[ \frac{\partial d_g^2(\mathbf{U}_i, \mathbf{U}_j)}{\partial \mathbf{U}_i} \right]_{ab} = \sum_{\ell=1}^r \frac{-2 \arccos \sigma_\ell}{\sqrt{1 - \sigma_\ell^2}} \frac{\partial \sigma_\ell}{\partial [\mathbf{U}_i]_{ab}} = \sum_{\ell=1}^r \frac{-2 \arccos \sigma_\ell}{\sqrt{1 - \sigma_\ell^2}} \mathbf{v}_{ij}^\ell \mathbf{w}_{ij}^{\ell\mathsf{T}},$$

where the first equality follows because $\sigma_\ell(\mathbf{U}_i^\mathsf{T} \mathbf{U}_j) = \sigma_\ell(\mathbf{U}_j^\mathsf{T} \mathbf{U}_i) = \sigma_\ell(\mathbf{U}_j \mathbf{U}_j^\mathsf{T} \mathbf{U}_i)$, and the second equality follows by parallel arguments as the derivation of (6) for the leading singular value. The last equation is the Euclidean gradient. Projecting onto the tangent space at $\mathbf{U}_i$, as described before Absil et al. (2009); Edelman et al. (1998), we obtain the following gradient on the Grassmannian

$$\boldsymbol{\nabla} d_g^2(\mathbf{U}_i, \mathbf{U}_j) = \sum_{\ell=1}^r \frac{-2 \arccos \sigma_\ell}{\sqrt{1 - \sigma_\ell^2}} (\mathbf{I} - \mathbf{U}_i \mathbf{U}_i^\mathsf{T}) \mathbf{v}_{ij}^\ell \mathbf{w}_{ij}^{\ell\mathsf{T}}.$$

## G  ACCELERATED LINE SEARCH ALGORITHM

Algorithm 1 outlines the Accelerated Line Search (ALS) given in Absil et al. (2009), used in the proof of Theorem 2.1.

---

**Algorithm 1** Accelerated Line Search (ALS)

---

**Require:** Riemannian manifold $\mathcal{M}$; continuously differentiable scalar field $f$ on $\mathcal{M}$; retraction $R$ from $T\mathcal{M}$ to $\mathcal{M}$; scalars $\eta_0 > 0$, $c, \beta, \gamma \in (0, 1)$.
**Input:** Initial iterate $\mathbb{U} \in \mathcal{M}$
**Output:** Sequence of iterates $\{\mathbb{U}_t\}$.
**for** $t = 0, 1, 2, \ldots$ **do**
    Pick $\boldsymbol{\Delta}_t \in T_{\mathbb{U}_t} \mathcal{M}$ such that the sequence of tangent vectors $\{\boldsymbol{\Delta}_t\}$ is gradient related.
    Select $\mathbb{U}_{t+1}$ such that

$$f(\mathbb{U}_t) - f(\mathbb{U}_{t+1}) \geq c(f(\mathbb{U}_t) - f(R_{\mathbb{U}_t}(\eta_t \boldsymbol{\Delta}_t))), \tag{7}$$

    where $\eta_t$ is the Armijo step size for the given $\eta_0, \beta, \gamma, \boldsymbol{\Delta}_t$.
**end for**

---

