# OpenReview forum: "Fusion over the Grassmannian for High-Rank Matrix Completion"
_ICLR.cc/2024/Conference — Submitted to ICLR 2024_

### Official Review · Reviewer_eDfV · 2023-10-19

**Soundness:** 2 fair
**Presentation:** 2 fair
**Contribution:** 1 poor
**Rating:** 3
**Confidence:** 3

**Summary:**

The paper introduces a fresh approach for clustering and filling in data distributed across multiple subspaces. It uses points in the Grassmannian as intermediaries and relies on the geometry of this Riemannian manifold for local convergence guarantees. Notably, this method doesn't require prior knowledge of the number of subspaces, can handle noisy data, and only needs an upper bound on the subspaces' dimensions. The paper also covers practical techniques for clustering, data completion, model selection, and sketching. Experiment results, both with synthetic and real data, indicate that this approach performs comparably to state-of-the-art methods in easy cases with high sampling rates and significantly surpasses them in challenging cases with low sampling rates, narrowing the gap towards the fundamental sampling limit of HRMC.

**Strengths:**

This paper stands out for its clarity and well-crafted presentation, ensuring readers can readily grasp the concepts and findings. Furthermore, it introduces a fresh and original formulation to address the challenge of HRMC (High-Rank Matrix Completion), offering a unique approach that departs from conventional methods.
The "related work" section is quite extensive, and the section "Fusion in practice" clarifies the method.

**Weaknesses:**

1) The paper discusses the HRMC problem but fails to provide a clear presentation or explanation of this problem. This omission makes it challenging for readers to understand the relevance and significance of the proposed model within the context of HRMC.

2) The section on Riemannian optimization lacks a clear demonstration of its utility. The presented concepts appear standard and do not introduce any novel insights. Additionally, widely available libraries like Pymanopt can automatically handle Riemannian gradients and appropriate retractions and line searches, making the inclusion of this optimization section less valuable. The complexity of the geodesic method used may not be practical, and it's rarely applied in practice. Furthermore, Theorem 2.1 seems to be a direct application of Proposition 4.7 from "An introduction to optimization on smooth manifolds" from Nicolas Boumal, which raises questions about the originality of this contribution.

3) The paper lacks transparency in explaining how hyperparameters are tuned, and the performance results shown in Figure 5 are not very promising. Overall, the numerical section is a little disappointing. These issues undermine the paper's practical applicability and potential for real-world impact.

**Questions:**

1) Why is it necessary to compute the gradient compared to automatic differentiation?
2) Can you explain the novelty in Theorem 2.1 compared to Proposition 4.7 from "An introduction to optimization on smooth manifolds" from Nicolas Boumal?
3) How are hyperparameters tuned in the numerical section?
4) Can you explain why the proposed method underperforms in Figure 5 on the Hopkins155 dataset?

---

### Official Review · Reviewer_hQSx · 2023-10-30

**Soundness:** 2 fair
**Presentation:** 2 fair
**Contribution:** 1 poor
**Rating:** 3
**Confidence:** 4

**Summary:**

The paper considers a high-rank matrix completion problem (HRMC), which is a well-studied combination of a matrix completion and a union of subspaces problem. The HRMC consists of assigning each column, of a given $m\times n$ matrix with missing entries, to a linear subspace, while minimising the number of such subspaces.

The main innovation of the reviewed work is the optimization problem formulated in eq (1) on pg. 2. The authors claim that HRMC can be well solved by an optimization over a product of a number of $n$ Grassmannian manifolds $\mathcal{G}(m,r)$, while minimizing the average distance of all the subspaces in respect to each other. The main theoretical statement is Theorem 2.1. on page 4, which is the consequence of the existing theory of Riemannian optimization to show convergence to a critical point, although the proof seems to be messy and use inconsistent notation. Although the paper at times does not use suitable terminology of Riemannian optimization, the derived update formula in eq (4) and above, is in fact a Riemannian gradient descent that exploits the known closed form expression for exponential retractions on Grassmannian manifold. The theoretical sampling limit is discussed on top of the page 6, and the authors cite the existing literature that sample complexity of HMRC is identical to matrix completion, but without stating what this means. The practical aspects of the algorithm are considered in Section 4, where the authors discuss the existing literature of clustering algorithms that could be applied to compute the number of subspaces. Section 5 provides numerical experiments on synthetic and real world datasets, which support that the optimization is an interesting concept, but do not achieve state-of-the-art results.

**Strengths:**

I find that this paper proposes an interesting idea in terms of a combining Riemannian optimization with clustering methods. The formulation of the optimization problem in eq (1) is novel. The paper provides experimental results and comparisons with other existing methods.

**Weaknesses:**

The paper suffers several weaknesses:
1) The terminology from Riemannian optimization is often incorrect or missing. For example it could be noted that eq (4) is the exponential map, or properly define Riemannian gradient with grad f(x). The proof of Theorem 2.1. does not seem to have consistent notation for the theorem. This is also seen in other parts of the paper, e.g. the notation of S and G for the Grasmannian manifold is inconsistent.
2) Although it is true that the Riemannian framework allows to show convergence to a critical point, the more interesting point is to investigate how the sample complexity affects the well-posedness of this formulation, and more importantly, the choice of the clustering algorithm for the n^2 matrix of distances.
3) There is very little practical or theoretical investigation of how the clustering algorithm affects overall performance.
4) The algorithm requires to compute distances between n^2 subspaces, making it computationally challenging.

Overall, I think the paper presents an interesting concept, but in its present form, it **does not** answer the interesting questions about it:
1) What are the situations for the proposed method to outperform the existing methods?
2) What are the requirments on the clustering method, and the number of subspaces, for the method to work well?
3) How to computationally deal with the n^2 matrix of distances? How can one effectively leverage this information?
4) What happens when the dimension of each subspace is large?

**Questions:**

1) Can you explain what happens when the rank is higher than the number of observations in a column, in other words r > \|\Omega_i\|, in this case I believe that d_c(x_i, U_i) == 0 and the largest principal angle is always going to be 0. In other words, there will be always intersecting lin. subspace between U_i and X_i.
2) Can you explain what is the proved convergence rate and how Theorem 2.1. goes beyond of just applying the existing results for Riemannian gradient descent?

---

### Official Review · Reviewer_2gRt · 2023-10-31

**Soundness:** 3 good
**Presentation:** 3 good
**Contribution:** 2 fair
**Rating:** 3
**Confidence:** 3

**Summary:**

This paper introduces an innovative approach for clustering and completing data distributed across a union of subspaces, harnessing points within the Grassmannian manifold. It leverages the intrinsic geometry of this Riemannian manifold to secure local convergence assurances.

This methodology can be seen as a natural extension of traditional convex clustering frameworks. Nevertheless, it deviates from convexity, leading to a local minimization challenge left unexplored within this paper.

Furthermore, the theoretical framework in this work is weak as the local convergence guarantee is a commonplace expectation for most optimization problems.

**Strengths:**

A new method and a new objective function is proposed; and a careful analysis of the local convergence guarantee.

**Weaknesses:**

The problem of local minimizer is not analyzed in the paper; and this should be a big issue as the objective function is highly nonconvex.

**Questions:**

Is there any guarantee that the algorithm converges to the underlying subspace?

---

### Official Review · Reviewer_H7XT · 2023-11-01

**Soundness:** 2 fair
**Presentation:** 3 good
**Contribution:** 2 fair
**Rating:** 3
**Confidence:** 3

**Summary:**

This work studies the problem of subspace recovery given $n$ partially observed sample points on a union of subspaces. The main idea is to first formulate the problem as an optimization on a Grassmann manifold, where the optimization variables are the $n$ subspaces corresponding to the $n$ points, and then to solve it by a gradient descent method. After obtaining estimation of the $n$ subspaces, one may use pairwise distance between every two subspaces to perform subspace clustering, and then do low-rank matrix completion within each cluster to recover the data points.

**Strengths:**

(1) The problems of subspace clustering and completion of data points on a union of subspaces are of broad interest.

(2) Numerical experiments on synthetic and real datasets demonstrate that the proposed algorithm can achieve better subspace clustering accuracy when the sampling rate is low.

(3) The paper is clearly written with ideas behind the problem formulation and algorithm well explained.

**Weaknesses:**

(1) The main contribution of this work is an algorithm that estimates the subspace $U_i$ corresponding to each data point $x_i$, and data completion and subspace clustering are extensions done by applying other algorithms on top of the estimation result. Thus, it would be more appropriate to emphasize subspace estimation instead of matrix completion in the title and abstract.

(2) The algorithm does not scale well as the number of points $n$ grows, because it assigns each $x_i$ a subspace $U_i$. Though it has been mentioned in the last paragraph in Section 4 that sketching can be adopted to reduce the number of samples, the proposed algorithm itself does not fully utilize the union of subspaces assumption to reduce complexity.

(3) The significance and novelty of this work could be enhanced by providing more theoretical analyses. For example, since the algorithm converges to a critical point of problem (1), it would be interesting to see what properties the critical points have.

**Questions:**

(1) What are the runtimes of the proposed algorithm in the numerical experiments? How efficient is it compared to the other algorithms?

(2) What is the subspace recovery accuracy in the synthetic data experiment, when comparing each estimated $U_i$ with $U_k^*$?

(3) How sensitive is the proposed algorithm to the choice of rank $r$ in the real data experiments, compared to other algorithms? Why fixing $r$ instead of tuning it as a hyper-parameter?

---

### Meta-Review · Area_Chair_amuz · 2023-12-05

**Metareview:**

This paper introduces a novel approach for clustering and completing data in a union of subspaces using Grassmannian points. It provides local convergence guarantees, works without prior knowledge of the number of subspaces, handles noise, and excels in low sampling rate scenarios, reducing the gap to the fundamental sampling limit of HRMC.

Four reviewers assessed the quality of the paper. They raised valid concerns about the scalability, novelty, and clarity of the paper. Unfortunately, the authors did not provide a response to these comments.

**Justification For Why Not Higher Score:**

The paper falls below the acceptance threshold for ICLR. The authors did not provide any response to the comments raised by the reviewers.

**Justification For Why Not Lower Score:**

N/A

---

### Decision · Program_Chairs · 2024-01-16

Reject